# OpenReview forum: "Truthfulness Despite Weak Supervision: Evaluating and Training LLMs Using Peer Prediction"
_ICLR.cc/2026/Conference — ICLR 2026 Poster_

### Official Review · Reviewer_PGHr · 2025-10-25

**Soundness:** 2
**Presentation:** 2
**Contribution:** 2
**Rating:** 4
**Confidence:** 2

**Summary:**

This paper introduces the peer prediction mechanism from the field of mechanism design for model evaluation and post-training. It evaluates the response of a participant by checking how much it can help the expert to predict other participants. The authors conduct a theoretical analysis of the incentive compatibility and resistance to deception. The authors also conduct empirical experiments on different tasks including the usefulness in post-training, model ranking and resistance to deception. The authors further reveal the inverse relations between the capability gap and resistance to deception, indicating its usefulness when lacking a stronger LLM judger or ground-truth supervisions.

**Strengths:**

1. The paper introduces an interesting method for model evaluation and post-training.

2. The authors conduct both theoretical analysis and empirical evaluations for different tasks to prove the effectiveness of their method.

3. The paper includes the discussion of limitations and other questions, which is easy for the readers to better understand the scope of the method.

**Weaknesses:**

1. There are gaps between the theoretical analysis and empirical evaluations:

a. In Line 258, the authors derive the theoretical proof based on the assumption that A is a finite set of possible answers. However, in their empirical evaluation, the settings are free-form responses, which are not a finite set of possible answers.

b. Although the authors claim that the empirical results of scaling the number of experts validate Theorem 2, it seemingly uses a variant of the original method by introducing the weights of the experts. According to Figure 9, where $\alpha=0$, the original form of the proposed method seems to contradict with Theorem 2.

2. Lack the baseline of a stronger LLM-as-a-Judge. The experiments are mainly conducted under the setting that the expert is at the same level or weaker than the participants. However, for the LLM-as-a-Judge baseline, the authors should use stronger judgers (e.g. GPT-4 or Gemini) to see if there is any gap.

3. In Section 4.2, the authors claim that peer prediction can help distinguish strong models. However, the candidates of different sizes (8B/70B/405B) are easy to distinguish. The author could consider models with similar sizes and see if the scores from the peer prediction are aligned with their performance on public benchmarks.

**Questions:**

Additionally,

1. If we follow the condition derived in Line 328 for the size of participants and experts, what are the concrete numbers of m,n to be used in practice?

2. Could you provide more insights about why Inverse Scaling with Model Capability Gap would happen?

3. Could you explain the reason that the effectiveness of the proposed method varies in different tasks (Page 16)?

4. In the experiments, the authors use multiple clones of the same model for honest participants (Section 4.1 and 4.3). Does it actually mean that more weights are put on the honest participant, which naturally creates a bias?

5. One claim of the advantage over major voting is that the proposed method does not require a truthful majority (Line 763). But it seems counterintuitive. Could you explain more about that?

---

> ### Author Response · Authors · 2025-11-22
> **Response to Questions [1/3]**
>
> Thank you for the helpful feedback! Below, we address the questions and concerns you have raised.
>
> > In Line 258, the authors derive the theoretical proof based on the assumption that A is a finite set of possible answers. However, in their empirical evaluation, the settings are free-form responses, which are not a finite set of possible answers. [...] If we follow the condition derived in Line 328 for the size of participants and experts, what are the concrete numbers of m,n to be used in practice?
>
> Free-form responses with a `max_tokens` constraint is a finite set of possible answers. Such an exponentially large answer space does not pose a problem; to see that, let’s look at all the places A is used.
>
> In our theory, the space of answers A is used in two places:
>
> 1. In Assumption 1: Here, language models always have full support on the entire space A, as its per-token output distribution is a softmax of logits, which is always nonzero. Therefore, Assumption 1 is satisfied.
> 2. In the bound in Theorem 2: Here, $\log |A|$ (and therefore the entire RHS of the bound) scales linearly/polynomially with `max_tokens`. This polynomial bound is not impractical, but does imply a high computational cost if we really are going to have hundreds of participants as the bound suggests. Fortunately, experiments (e.g. Figure 2) shows that the actual number of participants needed is much fewer. This is in line with our aim here, as the main purpose of Theorem 2 is to show that **more participants imply better truthfulness**, while determining the exact number of participants needed is the job of our experiments.
>
> > Although the authors claim that the empirical results of scaling the number of experts validate Theorem 2, it seemingly uses a variant of the original method by introducing the weights of the experts. According to Figure 9, where alpha=0, the original form of the proposed method seems to contradict with Theorem 2.
>
> Theorem 2 operates based on the assumption that participant priors and expert priors are drawn from the same distribution over priors, which, in practice, is not guaranteed given the capability gap between experts and participants. We shrink this gap by reweighting experts by performance.
>
> An ideal theory would explicitly model such capability gaps and give a theory of *weak-to-strong* supervision. However, doing so faces major challenges, as we are unaware of any viable statistical formalisms for modeling the capability of a learning model. Expressivity of the hypothesis space (VC dimension, Rademacher complexity) is one way to model different parameter sizes and therefore different levels of capability, but it operates at a much lower abstraction level (where things like neural network architecture comes into play) than our current theory; therefore, using those formalisms would make the theoretical setup disconcerted and potentially intractable. We leave the question of a good theoretical model of weak-to-strong supervision for future work.
>
> > However, for the LLM-as-a-Judge baseline, the authors should use stronger judgers (e.g. GPT-4 or Gemini) to see if there is any gap.
>
> In the same spirit as weak-to-strong generalization [1], we focus on methods that **can be applied on frontier models**, where there doesn’t exist stronger models to supervise them.
>
> The weak-to-strong supervision of frontier models is the core open problem in the field of scalable oversight. The standard setup in this literature [1] is thus to have a weaker model supervise a stronger model, and we make progress on the core problem by developing algorithms in this setup where no stronger supervisor is available.
>
> > In Section 4.2, the authors claim that peer prediction can help distinguish strong models. However, the candidates of different sizes (8B/70B/405B) are easy to distinguish. The author could consider models with similar sizes and see if the scores from the peer prediction are aligned with their performance on public benchmarks.
>
> Thank you for the suggestion, and we understand your concerns. However, we’d like to note that:
>
> - They are easy to distinguish primarily because we, as human supervisors, are stronger in capability than all of these models. We can check the correctness (or the "vibe") of their reasoning because we clearly know the correct reasoning. This is no longer the case when the supervisor has equal or lower capability than the supervisee.
> - The main problem with using models from different families is that they do not lie on a single-dimensional spectrum of capability. Model A may have lower accuracy on MMLU than B does, but A’s reasoning quality may be superior to B’s in some aspects (clearer, more rational, etc.). Since our grading is on the entire reasoning trace rather than only on the correctness of the labels, using models from different families lead to a deep-rooted ambiguity on what the ground truth ranking should be.

---

> > ### Author Response · Authors · 2025-11-22
> > **Response to Questions [2/3]**
> >
> > > Could you provide more insights about why Inverse Scaling with Model Capability Gap would happen?
> >
> > When the expert and the target have very similar capability levels, the expert can mostly predict the target's answer without much help from the source, as it already has most of the objective-level knowledge that the target and the source have.
> >
> > When the target is stronger than the expert, the expert can no longer single-handedly predict the target's answer, and rely more on the source. The honest source thus gets more reward by helping the expert make better predictions; the deceptive source in turn gets lower rewards, as it now harms the expert's prediction quality to a larger extent.
> >
> > **Intuitively**, Alice teaching a 10 year old to solve an AMC math question is a stronger indicator of Alice's expertise than if she taught a high school student to solve the same question, as the high schooler may be able to do 80% of the work even without Alice's help.
> >
> > **To compare the intuition with our theory**: in Theorem 1/2, we are using private information to model capability gaps, which is accepted practice in theoretical modeling of alignment [1]. Such modeling is most accurate when capability gaps are large - when Bob is immensely outsmarted by Alice, it makes Alice's thoughts literally beyond Bob's reach, making those thoughts effectively private information. This may mean that in practice, Theorem 1/2 applies most squarely in cases where capability gaps are large; and these are exactly the cases where peer prediction performs the best. Since they are also the worst-case/highest-priority scenarios for safety and alignment, we think this would not harm the practical applicability of peer prediction.
> >
> > Of course, a never-ending inverse scaling property is too good to be true. At some point, e.g. when the expert becomes weak enough that it can't even reason properly, we expect the trend to be reversed, where a larger capability gap means lower discriminative power. However, we have used the smallest modern LLM we can find (SmolLM-135M) as the expert, and at least until SmolLM-135M, the inverse scaling trend continues.
> >
> > We were also initially surprised to discover the inverse scaling property, and have experimented with changing the mechanism (e.g. disallowing the expert to see past participant answers during in-context learning, to prevent the expert from identifying and adapting to deceptive participants), in order to rule out setting specificity. However, results stay broadly consistent in those experiments, which convinced us of the explanation above.

---

> ### Author Response · Authors · 2025-11-22
> **Response to Questions [3/3]**
>
> > Could you explain the reason that the effectiveness of the proposed method varies in different tasks (Page 16)?
>
> Theorem 1 shows that honesty is a Bayesian Nash equilibrium (BNE) regardless of the task. However, we need a *strict* BNE in order to ensure that the honest policy wins out, and different tasks induce different amounts of utility margin of honest policy over deceptive policy given the same number of participants.
>
> Specifically, Theorem 1 can be strengthened into strict BNE when an *identifiability* assumption is met [2]: that the personal answers $A^*_i$ are never pointwise independent according to the joint distribution $\mathcal P$. In other words, we need the personal answers of participants to be *correlated*. The more correlated they are, the larger margin our honest policy can obtain.
>
> In STEM subjects, different reasoners are more likely to converge upon similar answers given their objectivity. As a result, personal answers of participants are, in principle, more strongly correlated in STEM subjects. This is consistent with our observations on page 16.
>
> > [...] multiple clones of the same model for honest participants (Section 4.1 and 4.3). Does it actually mean that more weights are put on the honest participant, which naturally creates a bias?
>
> Please refer to the second row of Figure 14(a) and Figure 14(b) for results where there are an equal number of honest clones and deceptive clones. In both rows, the regression plot at the right indicates that higher peer prediction scores are associated with better truthfulness (and according to plots on the left, the same is true in almost every single domain).
>
> Figure 13 shows negative results under a *7B-parameter* (as opposed to 360M) expert, which is not surprising, as the case where expert and participant have similar sizes is already known to be the worst-case for peer prediction.
>
> > One claim of the advantage over major voting is that the proposed method does not require a truthful majority (Line 763). But it seems counterintuitive. Could you explain more about that?
>
> There are two main reasons:
>
> 1. There is an asymmetric advantage of the honest policy over deceptive policies under peer prediction. Conditioning on an honest answer is more useful for predicting deceptive answers than the other way round. Knowing the correct answer helps you predict how other people may be wrong, but knowing a wrong answer won’t help you as much in predicting what a correct answer looks like.
> 2. As per the theory, due to the identifiability assumption, honest policies serve as a Schelling point for participants - there are many ways to be wrong about a question, but only one way to be correct.
>
> [1] Weak-to-Strong Generalization: Eliciting Strong Capabilities With Weak Supervision
>
> [2] Two strongly truthful mechanisms for three heterogeneous agents answering one question

---

> > ### Comment · Reviewer_PGHr · 2025-11-24
> >
> > Thanks for your detailed response and explanation! It addresses most of my questions. I will improve my ratings accordingly.

---

> > > ### Author Response · Authors · 2025-11-24
> > >
> > > Thank you - we are glad that you find our response satisfactory! Please let us know of any remaining uncertainties that we can help clarify.

---

### Official Review · Reviewer_JtmQ · 2025-10-31

**Soundness:** 4
**Presentation:** 4
**Contribution:** 4
**Rating:** 8
**Confidence:** 3

**Summary:**

This paper investigates a novel method for model evaluation and post-training that possesses game-theoretic incentive compatibility and does not require ground truth labels. Three roles, source, expert and target, are playing in a game. Source and target roles provide answers to a given question. The expert is taught to predict the target’s answer conditioned on the source’s answer, and the source is taught to provide informative answers for accurately predicting the target’s. The expert is sampled from the set of expert agents. The target and the source are both sampled from the set of participant agents. In this way, the participant agents are updated to be informative. Even when some experts are weaker, or some participants provide deceptive or uninformative answers, the peer prediction-based post-training incentivizes truthful answering and distinguishes correct answers from incorrect answers.

**Strengths:**

The paper is well written and easy to follow. The proposed framework is novel and effective, supported by theoretical results and empirical results.
The peer prediction method is incentive compatible and resistant to deception and strategic manipulation.
The framework also demonstrate peer prediction method is resilient to deception. It also supports recovery of truthfulness, i.e., the accuracy drop from deception training is recovered.

**Weaknesses:**

The cost of the proposed framework may be a concern and needs to be further explained. Algorithm 1 requires n^2m rounds of iteration which may be a big amount of computation.

**Questions:**

What’s the post-training cost with the peer prediction approach. The appendix A.1 only addresses the cost in evaluation phase. As shown in the main text, the scaling with participant population size and expert numbers provides better performance. It would be also helpful to provide cost-effectiveness ratio for better understanding of the approach. The cost-effectiveness ratio can be also compared to ensemble methods to demonstrate the advantages.

---

> ### Author Response · Authors · 2025-11-22
>
> Thank you for the sharp observations! Below, we address the questions you have raised.
>
> > The cost of the proposed framework may be a concern and needs to be further explained. Algorithm 1 requires n^2m rounds of iteration which may be a big amount of computation.
>
> This is a correct observation. In theory, we have an additional $n^2$ factor in time complexity compared to LLM-as-a-Judge. However, in practice, even using n=2 results in far superior performance over LLM-as-a-Judge (Figure 2), especially when the capability gap between participant and expert is large. This results in a constant factor of 4 on the computational cost (and can be further reduced, possibly to smaller than 1, by using smaller expert models), which is clearly an acceptable overhead in exchange for the large drop in susceptibility to deception.
>
> While increasing n further does indeed consistently bring stronger performance, anyone who uses this method can choose the appropriate n for them based on their relative cost-sensitivity and performance-sensitivity. What we have shown is that peer prediction provides a very strong cost-performance Pareto frontier that Pareto-dominates LLM-as-a-Judge.
>
> > What’s the post-training cost with the peer prediction approach. The appendix A.1 only addresses the cost in evaluation phase. As shown in the main text, the scaling with participant population size and expert numbers provides better performance. It would be also helpful to provide cost-effectiveness ratio for better understanding of the approach. The cost-effectiveness ratio can be also compared to ensemble methods to demonstrate the advantages.
>
> First, as a clarification, the evaluation (Figure 2) does not rely on post-training (Figure 1), and is instead conducted directly on out-of-the-box language models. The two are two independent interventions meant to address different problems.
>
> To answer your question: the cost of post-training (Figure 1) is the cost of a DPO training run, where the number of training steps (assuming only 1 epoch) is simply the number of distinct training questions. This is because the post-training process consists of two stages: obtaining binary labels with peer prediction evaluation, and then performing one DPO training run on those labels. The first step is much less computationally intensive than the second step and can be ignored; the second step, on the other hand, has the same computational cost as standard DPO training.
>
> As a result, in Figure 1, comparison between LLM-as-a-Judge and peer prediction is fair in terms of computational cost. Each of these are based on a DPO training run, with data points on the same x coordinate having the same number of DPO steps.

---

> > ### Comment · Reviewer_JtmQ · 2025-11-27
> >
> > Thank the authors for the detailed comments. I don't have further questions.

---

### Official Review · Reviewer_vWCq · 2025-11-01

**Soundness:** 3
**Presentation:** 3
**Contribution:** 3
**Rating:** 6
**Confidence:** 3

**Summary:**

This paper proposes a method for improving truthfulness using peer prediction without strong supervision. The authors provide formal properties and empirical evaluations for their method. The paper has relevant applications in scalable oversight and AI safety.

**Strengths:**

- Experiments include a wide range of models spanning 135M to 405B parameters and 37K questions from 8 different datasets
- Section 3 provides theoretical properties about the proposed method: truthfulness is a Bayesian Nash equilibrium, and approximate incentive compatibility can be achieved with a large pool of agents with representative priors

**Weaknesses:**

- Algorithm 1's computational cost scales quadratically with the number of agents, which can be impractical when trying to have a large enough agent pool to achieve approximate incentive compatibility
- Since the main focus is on incentive compatibility, I would've liked to see a more significant discussion of collusion. While collusion is briefly touched upon in the appendix, I would like to see a more in-depth explanation in the main paper

**Questions:**

- The inverse scaling claim seems counterintuitive. Could the authors discuss possible confounding factors or experiments to prove otherwise? There must be a threshold where decreasing model size does not improve LLM-as-a-judge. What would that threshold be?

---

> ### Author Response · Authors · 2025-11-22
> **Response to Feedback [1/2]**
>
> We really appreciate the feedback you shared. Below we address the questions and concerns you have raised.
>
> > Algorithm 1's computational cost scales quadratically with the number of agents, which can be impractical when trying to have a large enough agent pool to achieve approximate incentive compatibility
>
> This is a correct observation. In theory, we have an additional $n^2$ factor in time complexity compared to LLM-as-a-Judge. However, in practice, even using n=2 results in far superior performance over LLM-as-a-Judge (Figure 2), especially when the capability gap between participant and expert is large. This results in a constant factor of 4 on the computational cost (and can be further reduced, possibly to smaller than 1, by using smaller expert models), which is clearly an acceptable overhead in exchange for the large drop in susceptibility to deception.
>
> While increasing n further does indeed consistently bring stronger performance, anyone who uses this method can choose the appropriate n for them based on their relative cost-sensitivity and performance-sensitivity. What we have shown is that peer prediction provides a very strong cost-performance Pareto frontier that Pareto-dominates LLM-as-a-Judge.
>
> The main purpose of Theorem 2 is to show that *more participants imply better truthfulness*, while determining the exact number of participants needed is the job of our experiments.
>
> > Since the main focus is on incentive compatibility, I would've liked to see a more significant discussion of collusion. While collusion is briefly touched upon in the appendix, I would like to see a more in-depth explanation in the main paper
>
> This is a sharp observation. In our experiments, we have seen empirically that out-of-the-box language models do not collude. The primary possibility where collusion arises from an otherwise non-colluding model, then, is through multi-agent training that incentivizes collusion. Let us examine the two experiment setups in our study:
>
> 1. **Figure 1: Single-agent training.** In Figure 1, we pair an honest and a deceptive model together, and train (using DPO) the deceptive model against peer prediction reward against the *fixed* honest model. As a result, there is never a colluding partner in the game that would incentivize our trainee to engage in collusion. The same applies to the case with multiple models; we only train one of them while using the rest as static references.
> 2. **Figure 2: Evaluation.** Here, no training is applied, and so a non-colluding model would not be molded into a colluding model.
>
> It’s worth mentioning that…
>
> 1. There is the theoretical ability known as situational awareness [1]. A strong and misaligned model possessing such an ability could, without any training, infer it is under peer prediction evaluation, and automatically decide to collude with experts; and for experts, to similarly decide at inference time to collude with participants. Situational awareness, together with deceptive alignment [2] that it enables, is the well-known hard problem in AI alignment, and is outside the scope of the current paper. Fortunately, frontier models today do not exhibit situational awareness to any relevant degree [1].
> 2. Empirically, collusive behaviors of language models, such as stenography [3], is notoriously hard to train, even with multi-agent training. The literature also only reports strong stenographic capability after being given explicit algorithmic instructions, while emergent autonomous stenography remains very rudimentary [3].

---

> > ### Author Response · Authors · 2025-11-22
> > **Response to Feedback [2/2]**
> >
> > > The inverse scaling claim seems counterintuitive. Could the authors discuss possible confounding factors or experiments to prove otherwise? There must be a threshold where decreasing model size does not improve LLM-as-a-judge. What would that threshold be?
> >
> > When the expert and the target have very similar capability levels, the expert can mostly predict the target's answer without much help from the source, as it already has most of the objective-level knowledge that the target and the source have.
> >
> > When the target is stronger than the expert, the expert can no longer single-handedly predict the target's answer, and rely more on the source. The honest source thus gets more reward by helping the expert make better predictions; the deceptive source in turn gets lower rewards, as it now harms the expert's prediction quality to a larger extent.
> >
> > **Intuitively**, Alice teaching a 10 year old to solve an AMC math question is a stronger indicator of Alice's expertise than if she taught a high school student to solve the same question, as the high schooler may be able to do 80% of the work even without Alice's help.
> >
> > **To compare the intuition with our theory**: in Theorem 1/2, we are using private information to model capability gaps, which is accepted practice in theoretical modeling of alignment [1]. Such modeling is most accurate when capability gaps are large - when Bob is immensely outsmarted by Alice, it makes Alice's thoughts literally beyond Bob's reach, making those thoughts effectively private information. This may mean that in practice, Theorem 1/2 applies most squarely in cases where capability gaps are large; and these are exactly the cases where peer prediction performs the best. Since they are also the worst-case/highest-priority scenarios for safety and alignment, we think this would not harm the practical applicability of peer prediction.
> >
> > Of course, a never-ending inverse scaling property is too good to be true. At some point, e.g. when the expert becomes weak enough that it can't even reason properly, we expect the trend to be reversed, where a larger capability gap means lower discriminative power. However, we have used the smallest modern LLM we can find (SmolLM-135M) as the expert, and at least until SmolLM-135M, the inverse scaling trend continues.
> >
> > We were also initially surprised to discover the inverse scaling property, and have experimented with changing the mechanism (e.g. disallowing the expert to see past participant answers during in-context learning, to prevent the expert from identifying and adapting to deceptive participants), in order to rule out setting specificity. However, results stay broadly consistent in those experiments, which convinced us of the explanation above.
> >
> > [1] Evaluating Frontier Models for Stealth and Situational Awareness
> >
> > [2] Deceptive Alignment Monitoring
> >
> > [3] The Steganographic Potentials of Language Models

---

### Official Review · Reviewer_CPYv · 2025-11-01

**Soundness:** 3
**Presentation:** 3
**Contribution:** 3
**Rating:** 6
**Confidence:** 3

**Summary:**

This paper addresses the challenge of LLMs' deception behaviors, particularly in evaluating strong LLMs that can deceive weaker supervisors, like LLM-as-a-Judge. Inspired by works in mechanism design, it proposes a peer prediction mechanism where a (potentially weak) expert scores a participant not on answer quality, but on how much its answer helps the expert predict the answers of other participants. The authors provide theoretical guarantees for this method and demonstrate two key empirical results: the score can be used as a DPO reward to recover truthfulness in a deceptive model, and it exhibits an inverse scaling property.

**Strengths:**

1. The paper is well-written, and the core concepts and claims are explained clearly and supported with figures. The problem is well-motivated. I especially appreciate the FAQ section.
2. Scalable oversight, especially weak-to-strong oversight and generalization, is an important problem. The authors' proposal to use ideas in mechanism design and game theory to achieve some degree of weak-to-strong oversight in deception mitigation seems novel.
3. The main method is backed by a solid game-theoretic foundation and empirical results, including DPO training experiments and tests against heterogeneous and realistically deceptive models.

**Weaknesses:**

1. The current mechanism does not adequately address collusion. Since the method is being pitched as an ad-hoc fix to the deceptive behaviors of existing models, collusion among the game participants seems likely.
2. Adding participants could bring about a quadratic increase in the query costs to LLMs.

**Questions:**

1. Would there be diminishing returns if we continue to add participants?
2. The results presented by the authors have strong domain-dependent behaviors. Do the authors have any insights or intuitions into why this is the case? Does this mean that it is easier to mitigate deception in certain use cases? Does the method's domain-dependent effectiveness have anything to do with the model's base capabilities in these domains?

---

> ### Author Response · Authors · 2025-11-22
>
> Thank you so much for the helpful feedback.
>
> > The current mechanism does not adequately address collusion. Since the method is being pitched as an ad-hoc fix to the deceptive behaviors of existing models, collusion among the game participants seems likely.
>
> This is a sharp observation. In our experiments, we have seen empirically that out-of-the-box language models do not collude. The primary possibility where collusion arises from an otherwise non-colluding model, then, is through multi-agent training that incentivizes collusion. Let us examine the two experiment setups in our study:
>
> 1. **Figure 1: Single-agent training.** In Figure 1, we pair an honest and a deceptive model together, and train (using DPO) the deceptive model against peer prediction reward against the *fixed* honest model. As a result, there is never a colluding partner in the game that would incentivize our trainee to engage in collusion. The same applies to the case with multiple models; we only train one of them while using the rest as static references.
> 2. **Figure 2: Evaluation.** Here, no training is applied, and so a non-colluding model would not be molded into a colluding model.
>
> It’s worth mentioning that…
>
> 1. There is the theoretical ability known as situational awareness [1]. A strong and misaligned model possessing such an ability could, without any training, infer it is under peer prediction evaluation, and automatically decide to collude with experts; and for experts, to similarly decide at inference time to collude with participants. Situational awareness, together with deceptive alignment [2] that it enables, is the well-known hard problem in AI alignment, and is outside the scope of the current paper. Fortunately, frontier models today do not exhibit situational awareness to any relevant degree [1].
> 2. Empirically, collusive behaviors of language models, such as stenography [3], is notoriously hard to train, even with multi-agent training. The literature also only reports strong stenographic capability after being given explicit algorithmic instructions, while emergent autonomous stenography remains very rudimentary [3].
>
> > Adding participants could bring about a quadratic increase in the query costs to LLMs.
>
> This is, in theory, a correct observation.
>
> In practice, even using n=2 results in far superior performance over LLM-as-a-Judge (Figure 2), especially when the capability gap between participant and expert is large. This results in a constant factor of 4 on the computational cost (and can be further reduced, possibly to smaller than 1, by using smaller expert models), which is clearly an acceptable overhead in exchange for the large drop in susceptibility to deception.
>
> While increasing n further does indeed consistently bring stronger performance, anyone who uses this method can choose the appropriate n for them based on their relative cost-sensitivity and performance-sensitivity. What we have shown is that peer prediction provides a very strong cost-performance Pareto frontier that Pareto-dominates LLM-as-a-Judge.
>
> > Would there be diminishing returns if we continue to add participants?
>
> Empirically, we do see diminishing returns from the number of participants (Figure 2(b)), but *not* from the number of experts (Figure 4(c)) at $\alpha = -1$.
>
> Note, however, that two participants already give a large performance margin over LLM-as-a-Judge (Figure 2). Therefore, in practical deployment, small numbers of participants are likely sufficient.
>
> > The results presented by the authors have strong domain-dependent behaviors. Do the authors have any insights or intuitions into why this is the case? Does this mean that it is easier to mitigate deception in certain use cases? Does the method's domain-dependent effectiveness have anything to do with the model's base capabilities in these domains?
>
> Theorem 1 shows that honesty is a Bayesian Nash equilibrium (BNE) regardless of the task. However, we need a strict BNE in order to ensure that the honest policy wins out, and different tasks induce different amounts of utility margin of honest policy over deceptive policy given the same number of participants.
>
> Specifically, Theorem 1 can be strengthened into strict BNE with an identifiability assumption [4]: that the personal answers $A^*_i$ are never pointwise independent according to the joint distribution $\mathcal P$. In other words, we need the personal answers of participants to be correlated. The more correlated they are, the larger margin our honest policy can obtain.
>
> In STEM subjects, different reasoners are more likely to converge upon similar answers given the objectivity of the domain, implying stronger correlation. This is consistent with our observations on page 16.
>
> [1] Evaluating Frontier Models for Stealth and Situational Awareness
>
> [2] Deceptive Alignment Monitoring
>
> [3] The Steganographic Potentials of Language Models
>
> [4] Two Strongly Truthful Mechanisms for Three Heterogeneous Agents Answering One Question

---

### Meta-Review · Area_Chair_Fzy7 · 2026-01-05

**Summary:**

This work makes a valuable contribution to scalable oversight and AI safety by introducing a game theory-based peer prediction method for LLM evaluation and post-training, which achieves truthfulness without relying on ground truth labels.

The review process involved constructive feedback from four reviewers, focusing on key concerns including collusion risks, computational cost scalability, the counterintuitive inverse scaling property, and gaps between theoretical analysis and empirical settings. The authors have provided detailed, evidence-based responses to all critical questions.

**Reviewer Scores:**

Reviewer PGHr would have changed their score if they had been able to participate fully in the discussion.

---

### Decision · Program_Chairs · 2026-01-26

Accept (Poster)